# *Aspergillus nidulans* AmyG Functions as an Intracellular α-Amylase to Promote α-Glucan Synthesis

Alia Rizvi Syeda Kazim,[a] Yuting Jiang,[a] Shengnan Li,[b] (ID) Xiaoxiao He[a]

[a]Key Laboratory of Molecular Epigenetics, Ministry of Education, Institute of Genetics and Cytology, Northeast Normal University, Changchun, Jilin, China
[b]Jilin Institute of Biology, Changchun, Jilin, China

Alia Rizvi Syeda Kazim and Yuting Jiang contributed equally to this article. Author order was determined in order of increasing seniority.

**ABSTRACT** α-Glucan is a major cell wall component and a virulence and adhesion factor for fungal cells. However, the biosynthetic pathway of α-glucan was still unclear. α-Glucan was shown to be composed mainly of 1,3-glycosidically linked glucose, with trace amounts of 1,4-glycosidically linked glucose. Besides the α-glucan synthetases, amylase-like proteins were also important for α-glucan synthesis. In our previous work, we showed that *Aspergillus nidulans* AmyG was an intracellular protein and was crucial for the proper formation of α-glucan. In the present study, we expressed and purified AmyG in an *Escherichia coli* system. Enzymatic characterization found that AmyG mainly functioned as an α-amylase that degraded starch into maltose. AmyG also showed weak glucano-transferase activity. Most intriguingly, supplementation with maltose in shaken liquid medium could restore the α-glucan content and the phenotypic defect of a Δ*amyG* strain. These data suggested that AmyG functions mainly as an intracellular α-amylase to provide maltose during α-glucan synthesis in *A. nidulans*.

**IMPORTANCE** Short α-1,4-glucan was suggested as the primer structure for α-glucan synthesis. However, the exact structure and its source remain elusive. AmyG was essential to promote α-glucan synthesis and had a major impact on the structure of α-glucan in the cell wall. Data presented here revealed that AmyG belongs to the GH13_5 family and showed strong amylase function, digesting starch into maltose. Supplementation with maltose efficiently rescued the phenotypic defect and α-glucan deficiency in an Δ*amyG* strain but not in an Δ*agsB* strain. These results provide the first piece of evidence for the primer structure of α-glucan in fungal cells, although it might be specific to *A. nidulans*.

**KEYWORDS** *Aspergillus nidulans*, cell wall, α-glucan, AmyG, α-amylase

The fungal cell wall not only is responsible for cell morphology but also is required for other specialized functions, such as cell adhesion, pathogenesis, and drug resistance (1, 2). Among the components of the fungal cell wall, glucan is one of the major structural polysaccharides, consisting of glucose units. β-1,3/1,6-Glucan (here, β-glucan) is present in almost all fungal cell walls and is proinflammatory, while α-1,3-glucan (here, α-glucan) is present mainly in filamentous fungi, as well as some yeasts. Although α-glucan has no major role in fungal cell morphology (3, 4), it was suggested to form a protective layer (5, 6) and may mask β-glucan from the host immune system (7, 8). For instance, in *Histoplasma capsulatum* and *Magnaporthe oryzae*, α-glucan is the outside layer of the cell wall, which prevents the cells from being detected by the host immune response (7, 8); in *Aspergillus fumigatus*, loss of α-glucan caused remodeling of the cell wall structure and reduced fungal virulence (9). Apart from that, we and other groups found that α-glucan is critical for conidial adhesion (3, 4, 10). Artificial overexpression of α-glucan in *Aspergillus nidulans* could trigger cell wall reconstitution and thus promote fungal cell adhesion and biofilm formation (11).

Address correspondence to Shengnan Li, lishengnan8435@hotmail.com, or Xiaoxiao He, hexx100@nenu.edu.cn.

To date, the most detailed structural analysis of $\alpha$-glucan was performed in the *Schizosaccharomyces pombe* cell wall, which showed it as a long chain of $\alpha$-glucose units joined by 1,3-glycosidic linkages, with a short chain of 1,4-glycosidically linked $\alpha$-glucose at the reducing end (12). Short $\alpha$-1,4-glucan was also needed for $\alpha$-glucan synthesis in *Escherichia coli* (13). This $\alpha$-1,4-glucan was suggested to be the primer structure for $\alpha$-glucan synthesis, but it cannot be produced by $\alpha$-glucan synthetase itself (12). The primer structure of $\alpha$-glucan and the source of this oligosaccharide remain unclear. We previously characterized an amylase-like protein, AmyG, in *A. nidulans*. Deletion of AmyG significantly decreased the $\alpha$-glucan content in the cell wall (4, 10). Similar to AmyG, its homolog in *H. capsulatum* (Amy1) is proposed to be involved in $\alpha$-glucan synthesis (14).

Previous studies showed the effect of AmyG deletion on the cell wall of *A. nidulans*, but no *in vitro* characterization has been reported. In this study, we expressed AmyG in the *E. coli* system and performed biochemical assays to determine its function *in vitro*. We found that AmyG belongs to the glycoside hydrolase 13 (GH13) $\alpha$-amylase protein family. It had both hydrolytic and glucanotransferase activities. Supplementation with maltose or maltotriose in the growth medium could recover the conidial adhesion and $\alpha$-glucan content in a $\Delta amyG$ strain. These data suggested that AmyG provides $\alpha$-1,4-linked saccharides during $\alpha$-glucan synthesis in *A. nidulans*.

## RESULTS

**AmyG sequence alignment.** We previously reported that AmyG is an intracellular protein and is critical to promote $\alpha$-glucan synthesis in *A. nidulans* (4). According to previous sequencing analyses, AmyG shares sequence identity with a G6-forming $\alpha$-amylase of *Bacillus* sp. strain 707 (accession number NCBI:txid1416), and it was classified as a GH13 amylolytic enzyme (15). The only other $\alpha$-amylase-like protein that was reported to be important for $\alpha$-glucan synthesis was Amy1 in *H. capsulatum* (14). We further compared AmyG sequences with other known GH13 amylolytic enzymes. The phylogenetic tree showed that AmyG is clustered with members of the GH13_5 subfamily (Fig. 1), including *H. capsulatum* Amy1 and *Aspergillus niger* AmyD. Proteins of GH13 family members share similar structures and sequences at the catalytic sites (Carbohydrate-Active Enzymes database [www.cazy.org]). We compared the sequence of AmyG with those of other fungal amylases that were reported to affect the synthesis of cell wall $\alpha$-glucan, as well as a bacterial amylase. Although AmyG maintained most of the amino acids generally conserved in GH13 members, some sequences at the $(\alpha/\beta)$-barrel 7 were missing in AmyG (Table 1). In particular, the highly conserved aspartic acid, which was proposed to be critical for the catalytic reaction, was missing from this region (16).

**AmyG protein expression.** The *A. nidulans amyG* sequence was cloned into the pET28a plasmid to generate pET28a-*amyG*, in which AmyG was fused with a 6× His tag. The Rosetta strain was used as a host to express AmyG. The temperature and isopropyl-$\beta$-D-thiogalactopyranoside (IPTG) concentration were optimized for AmyG expression. Only trace amounts of AmyG were present in the supernatant fraction when cells were grown at 37°C (see Fig. S1A in the supplemental material), whereas substantial amounts of soluble AmyG were found in the supernatant fraction at 16°C (see Fig. S1B and C). IPTG concentrations ranging from 0.5 to 4 mM showed almost no differences in protein expression (Fig. 2A). Thus, the optimized conditions of 16°C and 1 mM IPTG were chosen for protein expression in the following experiment. The supernatant of the cell lysate was applied for affinity chromatography using a prepacked nickel-nitrilotriacetic acid (Ni-NTA) column. The yield of AmyG was tested by SDS-PAGE, which showed a specific band at around 72 kDa (Fig. 2B). The purified AmyG was confirmed by blotting with an anti-His antibody (Fig. 2C). The protein was then concentrated and resuspended in phosphate-buffered saline (PBS) (pH 7.6) by using an Amicon Ultra-15 centrifuge column with a 10-kDa cutoff value.

**Enzymatic characterization of AmyG.** According to the sequence analysis, AmyG was expected to be a GH13_5 family member. Fungal members in the GH13_5 family were reported to show hydrolytic activity (EC 3.2.1) (17) and glucanotransferase activity (EC 2.4.1) (18), and both proteins are important for fungal cell wall synthesis. Therefore, AmyG was tested in hydrolysis and transglycosylation assays.

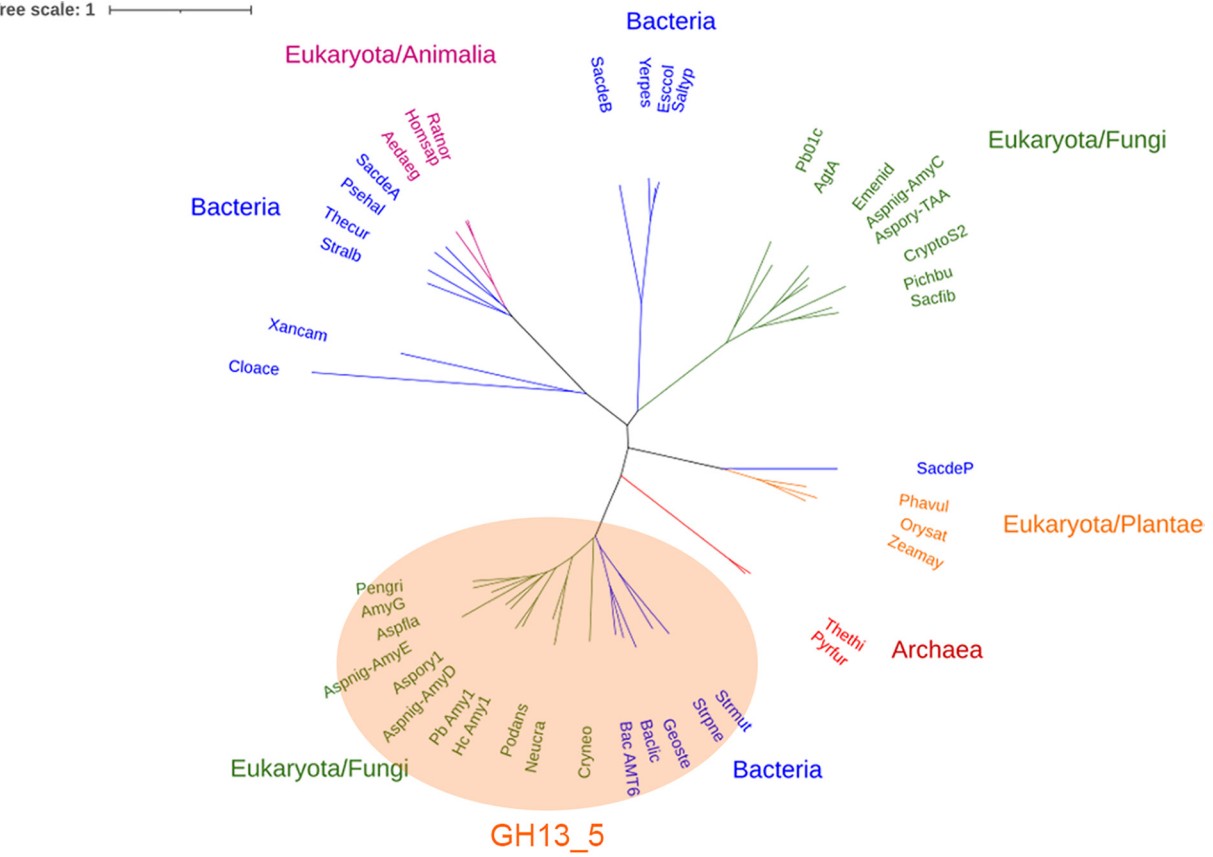

**FIG 1** Phylogenetic tree of $\alpha$-amylase proteins from representative taxa. Sequences from different domains/kingdoms are labeled with different colors. Full sequences of the $\alpha$-amylases are listed in Table S1 in the supplemental material.

AmyG was first incubated with starch to test its hydrolytic ability. AmyG mainly digested starch into glucose, maltose, and maltotriose, and small amounts of longer oligosaccharides were also formed during the reaction (3 h) (Fig. 3A). A longer incubation time (6 h) did not significantly alter the formation of these products (Fig. 3A). The variations in temperature (20°C to 37°C) and pH (4.5 to 8.5) had negligible effects on AmyG hydrolytic ability (see Fig. S2A and B), indicating that the function of AmyG was quite stable. In comparison, the amylase from *Aspergillus oryzae* mainly hydrolyzed the starch into glucose and maltose, with the formation of maltotriose and longer oligosaccharides being scarce (see Fig. S3A). We used the bicinchoninic acid (BCA) assay to quantify the reducing ends in the hydrolytic reaction mixture. The AmyG activity on starch was $67.4 \pm 7.3$ $\mu$mol reducing ends mg$^{-1}$ min$^{-1}$. The $K_m$ for starch was around 0.06% to 0.08% (wt/vol).

We then incubated AmyG with maltoheptaose (as acceptor) and maltose and maltotriose each (as donor) to assay the glucanotransferase activity. Instead of chain elongation as

**TABLE 1** Alignment of the catalytic conserved regions in $\alpha$-amylase

| Amylase | Sequence[a] | | | |
|---|---|---|---|---|
| | $\beta3$ | $\beta4$ | $\beta5$ | $\beta7$ |
| AmyG | **D**AVLN**H** | GM**RLD**AAKH | G**E**YW | – – – – – |
| *Aspergillus niger* AmyD | **D**AVLN**H** | GM**RLD**AVKH | G**E**YW | HSTNI**D** |
| *Histoplasma capsulatum* Amy1 | **D**TVLN**H** | GL**RLD**AAKH | A**E**YW | – – MN**HD** |
| *Paracoccidioides brasiliensis* Amy1 | **D**AVLN**H** | GL**RFD**AAKH | A**E**YW | – – MN**HD** |
| *Aspergillus niger* AgtA | **D**TVINN | GL**RID**AAKH | G**E**VL | FSEN**HD** |
| *Bacillus* AMT6 | **D**VVMN**H** | GF**RID**AVKH | A**E**FW | – – DN**HD** |

[a]Sequences corresponding to the $(\alpha/\beta)$ barrels 3, 4, 5, and 7 are listed. The generally conserved amino acid residues are shown in bold. Full sequences are listed in Table S1 in the supplemental material.

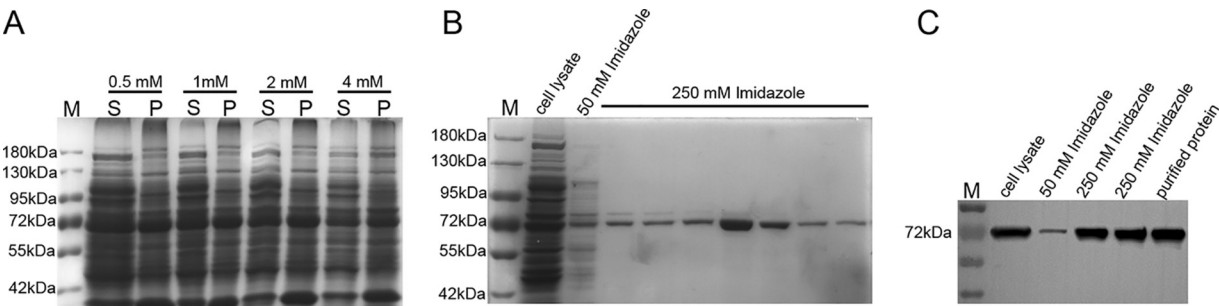

**FIG 2** Protein expression and purification of AmyG. (A) SDS-PAGE analysis of AmyG expression at 16°C with various IPTG concentrations. M, marker; S, supernatant fraction; P, pellet fraction. (B) SDS-PAGE analysis of AmyG purification. Samples from cell lysates (before purification), the washing step, and the elution step were tested. (C) Western blot analysis of AmyG expression. The indicated samples were blotted with an anti-His antibody (AE003; ABclonal, China).

we had expected, maltoheptaose was completely digested to smaller units of carbohydrates (Fig. 3B). We also tried to prolong the incubation time to 12 h, since the glucanotransferase activity normally started with hydrolysis. However, the same results were found for those assays, except that fewer longer oligosaccharides were present (Fig. 3B). Intriguingly, we found that the maltotriose we added in the reaction mixture was also hydrolyzed, because no maltotriose was accumulated in the final products.

Next, we tested AmyG with maltoheptaose, maltose, maltotriose, and glycogen individually. As expected, AmyG hydrolyzed maltoheptaose into smaller saccharides and had no effect on maltose and glycogen (Fig. 3C). When AmyG was incubated with only maltotriose, not only maltose but also maltotetraose and even longer oligosaccharides were present in the reaction mixture (Fig. 3C), indicating that AmyG also had glucanotransferase activity. The band of maltotetraose was also found in the reaction of maltotriose plus glucose but not in the reaction of maltose plus glucose (see Fig. S4), indicating that AmyG uses only maltotriose and glucose for the glucanotransferase reaction. In addition, the maltotetraose produced was maintained at very low levels even when the reaction was performed overnight (see Fig. S3B), while maltose was always the major final product. Thus, these data suggested that AmyG has strong hydrolytic activity but very weak glucanotransferase activity.

Because thin-layer chromatography (TLC) can separate carbohydrates only by their relative sizes, we also used high-performance anion-exchange chromatography (HPAEC) to identify the products from the AmyG starch and maltotriose hydrolysis assays. Results were

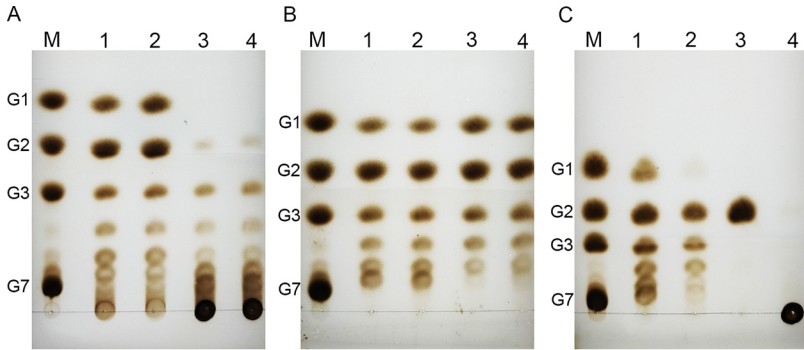

**FIG 3** TLC analyses of AmyG reaction products from hydrolysis and transglycosylation assays. (A) The hydrolysis reactions were set up by incubating starch with purified AmyG for 3 h (sample 1) or 6 h (sample 2) at 37°C. The same reactions were also set up by using heat-killed AmyG (sample 3) or the cell lysate of non-AmyG-expressing *E. coli* cells (sample 4) for 3 h at 37°C. (B) The glucanotransferase reactions were set up by incubating maltose plus maltoheptaose (sample 1) or maltotriose plus maltoheptaose (sample 2) with purified AmyG for 3 h at 37°C. The same reactions with maltose plus maltoheptaose (sample 3) or maltotriose plus maltoheptaose (sample 4) were also performed for 12 h at 37°C. (C) The hydrolysis reactions were set up by incubating AmyG with maltoheptaose (sample 1), maltotriose (sample 2), maltose (sample 3), or glycogen (sample 4) for 3 h at 37°C. For this figure, glucose (G1), maltose (G2), maltotriose (G3), and maltoheptaose (G7) were loaded as markers.

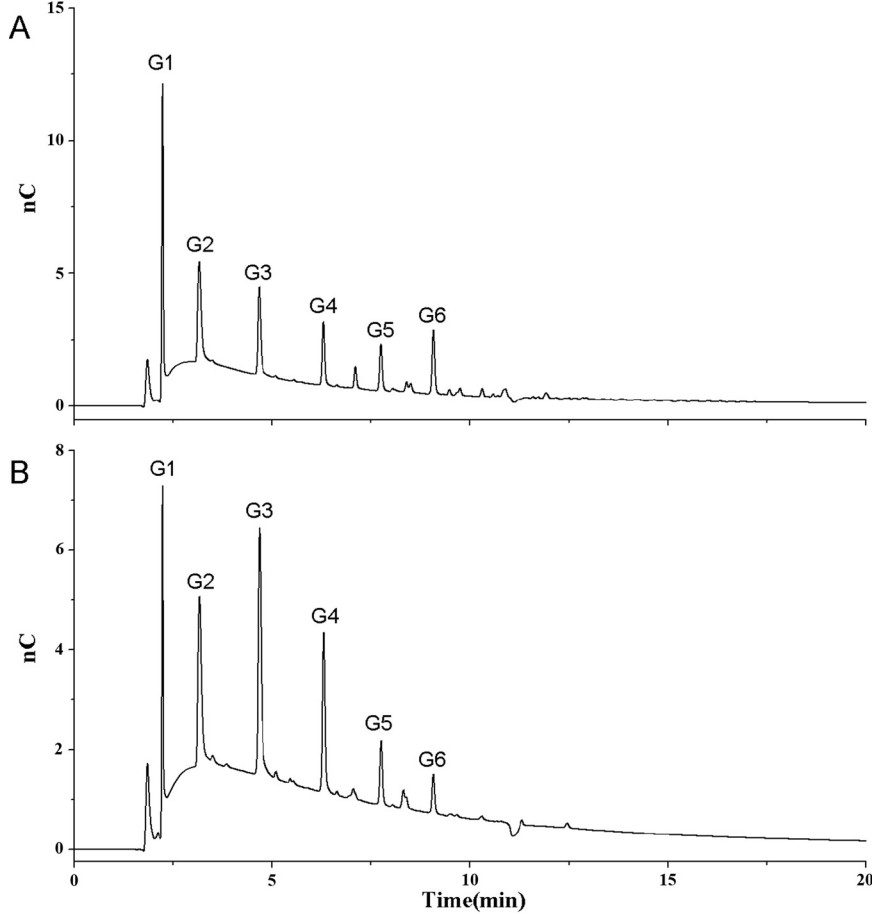

**FIG 4** HPAEC analyses of reaction products from AmyG starch or maltotriose hydrolysis assays. The hydrolysis reactions were set up by incubating purified AmyG with starch (A) or maltotriose (B) for 3 h at 37°C. The size of the product represented by each peak was determined by using standard chemicals (as shown in Fig. S4 in the supplemental material) and is indicated at the top of each peak (G1, glucose; G2, maltose; G3, maltotriose; G4, maltotetraose; G5, maltopentaose; G6, maltohexaose).

similar to data from TLC. In the AmyG starch hydrolysis reaction mixture, the final products included glucose, maltose, and G3 to G6 maltooligosaccharides; the same results were also obtained for the AmyG maltotriose hydrolysis assay (Fig. 4). Altogether, these data suggested that AmyG mainly functions as an amylase, producing maltose as the major product.

**Addition of maltose rescues the loss of AmyG in *Aspergillus nidulans*.** Since AmyG is responsible for the degradation of starch into maltose in *A. nidulans*, we wondered how it relates to the α-glucan synthesis pathway. Could it be the primer structure for α-glucan synthesis? If so, supplementation with maltose in liquid medium should be able to compensate for the loss of AmyG in *A. nidulans* cells.

The most distinguishable phenotype of α-glucan-defective *A. nidulans* strains, such as Δ*amyG* and Δ*agsB* strains, involved dispersed and tiny colonies in shaken liquid medium (Fig. 5A) (4, 10). Based on this phenotype, we grew the Δ*amyG* strain in liquid complete medium (CM) or CM supplemented with 0.5% maltose. The wild-type A1149 strain was used as a positive control, and the Δ*agsB* strain was used as a negative control. When grown in CM plus 0.5% maltose, the Δ*amyG* strain formed colony pellets similar to those of A1149, while the Δ*agsB* strain still formed tiny colonies as in regular CM (Fig. 5A). To test whether other oligosaccharides could rescue such phenotypic defects, we also tested the Δ*amyG* and Δ*agsB* strains in CM plus 0.5% maltotriose. The colony formation of the Δ*amyG* strain was partially rescued by supplementation with maltotriose, and no improving effect was observed for the Δ*agsB* strain (Fig. 5A). Since the rescue of colony formation was also related to osmotic balance, we further tried to rescue the Δ*amyG* strain with sorbitol, sucrose, and extra glucose.

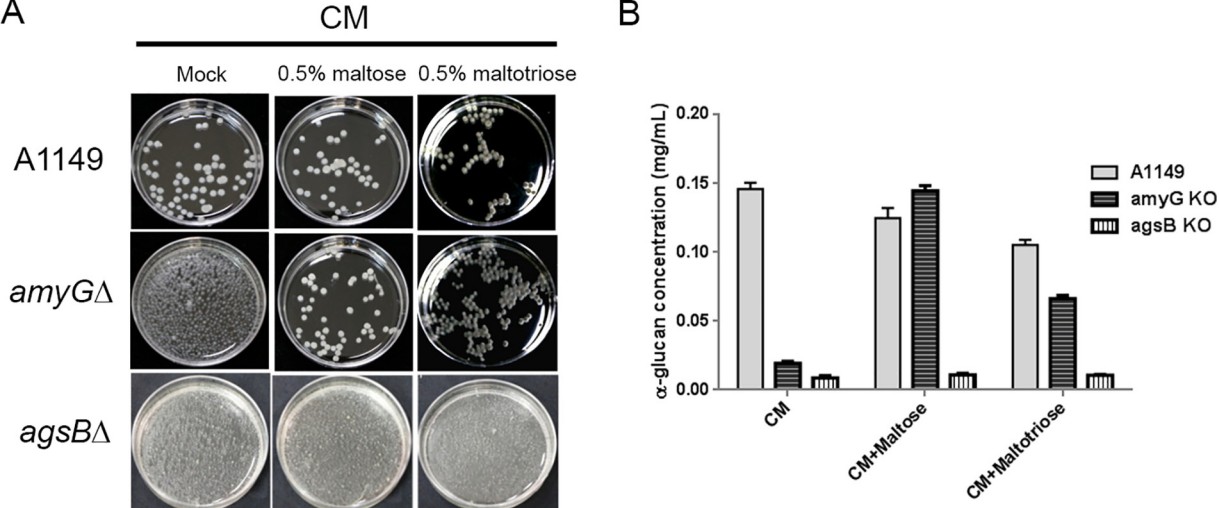

**FIG 5** Maltose and maltotriose rescued the phenotypic defect of the Δ*amyG* strain in shaken liquid medium. (A) Spores from the A1149, Δ*amyG*, and Δ*agsB* strains were inoculated in the indicated medium for 24 h at 150 rpm and 28°C. Typical colonies from each medium were transferred to a petri dish for imaging. (B) Quantification of α-glucan in the cell walls of the A1149, Δ*amyG*, and Δ*agsB* strains. Spores from each strain were inoculated in the indicated medium for 24 h at 150 rpm and 28°C. Glucose concentrations in the alkaline-soluble fraction were quantified by the anthrone assay.

However, none of them rescued the dispersed and tiny colonies in the shaken liquid medium (see Fig. S5), suggesting that the addition of maltose or maltotriose could directly recover the α-glucan content in the Δ*amyG* strain. To test that notion, colonies of the A1149 and Δ*amyG* strains that had been grown in different media were subjected to α-glucan quantification. Results showed that the addition of maltose almost completely restored the α-glucan content in the Δ*amyG* strain, while the addition of maltotriose only partially restored the α-glucan content (Fig. 5B). Moreover, extra maltotriose in the medium also showed an inhibitory effect on the α-glucan synthesis by the wild-type strain (Fig. 5B), which was also represented by smaller colony sizes in shaken liquid medium (Fig. 5A). In contrast, the addition of maltose or maltotriose had no rescue effect on the Δ*agsB* strain (Fig. 5B). Thus, the quantification results directly correlated with the colony formation of α-glucan-defective strains in different media.

## DISCUSSION

Although α-glucan is important for cell adhesion and host-microbial interactions, the structure and biosynthetic pathway of this cell wall component were not quite clear. The structure of *S. pombe* α-glucan suggested that it is a long chain of α-glucose with no branches (12). The α-glucan synthetase itself can only elongate an existing chain of glucan; it cannot synthesize it from the very beginning (12). For these reasons, other enzymes are needed to provide α-1,4-glucan as the primer structure. However, the exact structure and the source of this primer structure are still unknown.

To date, at least two amylase-like proteins have been shown to be essential for the α-glucan content in fungal cell walls (4, 10, 14), and several other amylase-like proteins were found to be relevant (17–20). Based on previous reports, *A. nidulans* AmyG not only was essential for α-glucan synthesis (4, 10) but also affected the structure of cell wall α-glucan (10). We analyzed the enzymatic function of AmyG in this study. Sequence analysis showed that AmyG belongs to the GH 13_5 subfamily, and it maintained most conserved amino acids for an amylase. The purified AmyG showed strong starch-hydrolyzing ability and weak glucanotransferase activity. The major product of AmyG was always maltose in our tests, which was due to its hydrolytic function. The glucanotransferase function of AmyG could be seen only in reaction mixtures containing maltotriose, and the final products of this reaction were G3 to G6 oligosaccharides. However, these oligosaccharides were always maintained at very low levels. The homolog of AmyG in *A. niger* could also form trace amounts of G4 to G7 oligosaccharides in starch hydrolysis reactions (17), indicating that this special intracellular amylase may have

additional function. Amylases with these two enzymatic functions were also reported in other cells (21).

If AmyG is the beginning step in the α-glucan synthesis pathway and its catalytic reaction produces maltose as the major product, it could be surmised that maltose could be the initial substrate for α-glucan synthesis. We found that both the phenotypic and α-glucan content defects of the ΔamyG strain could be compensated by the addition of maltose in the CM. In contrast, the addition of maltose failed to rescue such defects in the ΔagsB strain. Thus, these data suggested that maltose might be the primer structure for α-glucan synthesis. However, AmyG also produced G3 to G6 oligosaccharides, which also might be primer structures for α-glucan synthesis. Due to the lack of G4 to G6 oligosaccharides in large amounts, we used only maltotriose to further compensate for the deletion of AmyG. Maltotriose showed less efficiency to compensate for the loss of AmyG, which might be due to the uptake efficiency; it would be easier for the smaller maltose to enter the fungal cells from the medium. In addition, maltotriose itself showed an obvious inhibitory effect on α-glucan synthesis in the wild-type strain. Thus, the extra maltotriose in the medium might trigger a negative-feedback pathway to affect α-glucan synthesis.

In another study, Camacho and colleagues (19) used a G4/G5-oligosaccharide-producing enzyme (Amy1p) from *Paracoccidioides brasiliensis* to compensate for the loss of Amy1 in *H. capsulatum*. Although oligosaccharides were not directly tested in their assay, the results confirmed that the short-chain oligosaccharides were essential for α-glucan synthesis. The primer structure (α-1,4-glucan) reported by Grun et al. (12) was much longer than G2/G3 or G4/G5 maltooligosaccharides. This could indicate that the primer structures for α-glucan are not identical in different fungal cells. In addition, the sequences at the catalytic sites were not the same between AmyG and Amy1p (*P. brasiliensis*), which could explain the different compounds resulting from their hydrolytic reactions. Alternatively, an intermediary protein may further modify the maltose and maltotriose to longer oligosaccharides, as reported by Grun et al. (12). The localization of AmyG in *A. nidulans* cells showed no preference for the cell membrane (4), which did not favor the utilization of its products for cell wall synthesis. Thus, an additional protein might be involved in catalyzing maltose/maltotriose in *A. nidulans* cells. Further study is required to test whether other enzymes are also included in α-glucan synthesis. In conclusion, data from this study provided the first evidence that the source of maltose/maltotriose, which is provided by an intracellular amylase, is essential for α-glucan synthesis in *A. nidulans*.

## MATERIALS AND METHODS

***Aspergillus nidulans* strains and growth condition.** The *A. nidulans* A1149, ΔamyG, and ΔagsB strains were obtained from laboratory stock, as reported previously (4). All strains were grown on CM (1% glucose, 0.2% peptone, 0.1% yeast extract, 0.1% Casamino Acids, 50 ml 20× nitrate salts, 1 ml trace elements, 1 ml vitamin solution [pH 6.5]) supplemented with nutrition solution as required. Trace elements (2.2 g $ZnSO_4$·$7H_2O$, 1.1 g $H_3BO_3$, 0.5 g $MnCl_2$·$4H_2O$, 0.5 g $FeSO_4$·$7H_2O$, 0.17 g $CoCl_2$·$6H_2O$, 0.16 g $CuSO_4$·$5H_2O$, 0.15 g $Na_2MoO_4$·$2H_2O$, and 5 g $Na_4EDTA$ in 100 ml water [pH adjusted to 6.5 by KOH pellet]), vitamin solution (100 mg each of biotin, pyridoxine, thiamine, riboflavin, *p*-aminobenzoic acid [PABA], and nicotinic acid per 100 ml water), and nitrate salt stock solution (120 g $NaNO_3$, 10.4 g KCl, 10.4 g $MgSO_4$·$7H_2O$, 30.4 g $KH_2PO_4$ in 1 L water) were used as noted. Nutrition stocks used in this study included pyridoxine (50 mg/liter), uridine (1.2 g/liter), and uracil (1.12 g/liter). Details of all solution were described by Kaminskyj (22).

**Sequence analysis.** The gene sequences of *amyG* were attained from the *Aspergillus* Genome Database (http://www.aspgd.org/), with accession number AN3309. Other GH13 family member sequences were retrieved from GenBank and are all listed in Table S1 in the supplemental material. A phylogenetic analysis was performed by comparing the AmyG sequence against 42 amylase sequences. The phylogenetic tree was constructed by MEGA software using a neighbor-joining method (23).

**Cloning procedure.** Primers were designed using Invitrogen online software (AmyGF, CATATGATGTTGT CGCTCCTAACATGC; AmyGR, CTCGAGTTAGATAGCGTGGTAAATGTTCACA), and restriction enzyme sites for NdeI and XhoI were added. Concurrently, mRNA was extracted from hyphae of the A1149 strain by using the Omega fungal RNA kit. The TranScript One-Step genomic DNA (gDNA) removal and cDNA synthesis kit (TransGene, China) was used to generate cDNA, and the anchored oligo(dT)$_{18}$ primer from the kit was used in the reverse transcription reaction. The *amyG* sequence was amplified using a Platinum Pfx kit (Invitrogen). The sequences obtained were cleaned with an M012 PCR purification kit (Liaoning Neogene Biochemical Technology Co. Ltd.), cloned into pEASY using a blunt-end cloning kit, and eventually subcloned into pET-28a. All sequences were verified by Sanger sequencing. The bacterial strain DH5α was used during cloning, and the bacterial strain BL21 was used later for protein expression.

**Protein expression and extraction.** The plasmid pET28a-*amyG* was extracted from DH5$\alpha$ cells and transformed into the BL21 strain. Optimal growth conditions such as temperature (16°C and 37°C) and IPTG concentrations (0.5 mM, 1 mM, 2 mM, and 4 mM) were screened in parallel experiments. The yield of AmyG protein was tested by SDS-PAGE.

Protein purification was carried out with a Ni-NTA column (Qiagen), with different concentrations of imidazole for the washing and elution steps, i.e., 20 mM and 50 mM concentrations were used for washing and 250 mM was used for protein elution. After the purification, AmyG was concentrated by using an Amicon Ultra-15 centrifuge column with a 10-kDa cutoff value (Merck Millipore, Ltd.), and the suspension buffer was changed from 250 mM imidazole to PBS (pH 7.4). The purified protein was used within 1 week or stored in 25% glycerol in PBS (pH 7.4) at −80°C.

**Enzymatic assays.** Enzymatic characterization was performed by testing the glucanotransferase and hydrolytic abilities of AmyG protein. The standard chemicals sodium barbital, maltose, maltotriose, maltoheptaose, starch, and $\alpha$-amylase (Aladdin Chemical Retailers) were used.

The standard chemical reaction of the hydrolytic assay was performed in a total of 100 $\mu$l containing 20% starch (from a stock solution of 2% [wt/vol]), 75% 0.02 M sodium barbital buffer (pH 6.5), and 5 $\mu$l of fresh AmyG. Normally, 5 $\mu$l of AmyG contains 3 $\mu$g purified enzyme, representing 0.2 U (1 U was defined as the amount of enzyme needed to produce 1 $\mu$mol reducing ends min$^{-1}$). The reaction mixture was incubated at 37°C for 3 h, 6 h, or 12 h before further analysis. Different pH values and temperatures were tested for the AmyG starch-hydrolyzing ability. The maltotriose and maltoheptaose hydrolysis assays were also performed in the aforementioned system, except that maltotriose and maltoheptaose stock solutions (both 1% [wt/vol]) were used. For quantitative analysis of AmyG hydrolysis, the total reaction mixture was 500 $\mu$l (20% starch, 75% 0.02 M sodium barbital buffer, and 25 $\mu$l AmyG), and 50-$\mu$l samples were taken from the reaction mixture every 5 min. The reducing sugar was quantified by the BCA method (24). The $K_m$ value of AmyG for starch was tested by measuring AmyG with five different starch concentrations (ranging from 0.2% to 0.01% [wt/vol]). Each quantification experiment was performed three times in duplicate.

The chemical reaction of the glucanotransferase assay was performed in a total of 100 $\mu$l containing 10% maltoheptaose (stock of 1% [wt/vol]), 10% maltose (stock of 1% [wt/vol]) or maltotriose (stock of 1% [wt/vol]), 75% 0.02 M sodium barbital buffer (pH 6.5) with 5 $\mu$l of fresh AmyG. The same reaction was also performed for maltose or maltotriose with glucose (stock of 1% [wt/vol]). The reaction mixture was incubated at 37°C for 3 h or 12 h before further analysis.

The results of the hydrolysis and transglycosylation assays were detected by spotting 3 to 5 $\mu$l of the reaction mixture on TLC plates (silica gel 60 F$_{254}$; Thermo Fisher Scientific). After drying, the TLC plates were run in a small amount of running buffer (butanol/acetic acid/water, 2:1:1). Then the dried plates were submerged in 2% sulfuric acid (in ethanol) for 2 s and developed at 110°C for an appropriate period.

The results of the hydrolysis assay were also tested by HPAEC with a pulsed amperometric detector. A Thermo Fisher Scientific Dionex ICS-5000 system with a CarboPac PA200 column (4 $\mu$m; 3 by 250 mm) was used for detection. Standard chemicals of glucose, maltose, maltotriose, maltotetraose, and maltopentaose were mixed at 20 $\mu$g/ml to test the standard elution times.

**$\alpha$-Glucan quantification.** For $\alpha$-glucan quantification, $10^7$ spores from each strain were grown at 30°C in 100 ml of the indicated medium, shaken at 150 rpm for 24 h. The spherical colonies were collected on a Whatman filter paper, consecutively washed with double-distilled water (ddH$_2$O) and 0.5 M NaCl, and then placed at −80°C overnight. Colonies were washed with ddH$_2$O and 0.5 M NaCl and suspended in disruption buffer (80 mM Tris, 200 mM EDTA [pH 8]); an ultrasonic system (model JY98-IIIN; Jinxin Co. Ltd., China) was used to break down the colonies (power, 300 W; 15 s on and 20 s off, repeated for 20 cycles), and then the suspension was centrifuged at 3,500 × $g$ for 10 min to collect the cell wall fraction. The cell wall fraction was washed once with disruption buffer at 4°C for 2 to 4 h and then with ddH$_2$O at 4°C for 4 h. The cell wall pellet obtained after washing was lyophilized overnight. Subsequently, dry cell wall was suspended in 1 M NaOH (1 mg/ml) for 1 h at 65°C or overnight at 37°C. Next, the suspension was centrifuged at 12,000 × $g$ for 10 min, and the supernatant was separated. The pH was neutralized by adding acetic acid, and the suspension was again centrifuged at 12,000 × $g$ for 10 min; a transparent pellet was obtained, which was washed once with ddH$_2$O and incubated for 1 h at 100°C in 3 M H$_2$SO$_4$ (vortex-mixed thoroughly to dissolve glucan in the 3 M H$_2$SO$_4$). The samples were then used in the anthrone assay.

The anthrone assay was used to quantify $\alpha$-glucan. The anthrone reagent (2 mg/ml in chilled H$_2$SO$_4$) and samples were prepared on the same day, 100 $\mu$l of $\alpha$-glucan suspended in 3 M H$_2$SO$_4$ and 1 ml of anthrone reagent were mixed in a 1.5-ml Eppendorf tube, and the mixture was maintained at 100°C for 10 min. The tubes were cooled on ice, and readings were taken at 630 nm.

**Statistical analysis.** All $\alpha$-glucan quantification analyses were performed in three independent tests with duplicates each time. Histograms were created with GraphPad Prism v8 (GraphPad Software, La Jolla, CA, USA).

## SUPPLEMENTAL MATERIAL

Supplemental material is available online only.

**SUPPLEMENTAL FILE 1**, PDF file, 0.4 MB.
**SUPPLEMENTAL FILE 2**, XLSX file, 0.03 MB.

## ACKNOWLEDGMENTS

We thank Susan Kaminskyj (University of Saskatchewan, Saskatoon, Canada) for critical discussion of the manuscript.

This work was supported by the National Natural Science Foundation of China (grant 32070809) and the Fundamental Research Funds for the Central Universities of China (grant 2412019FZ025).

Author contributions were as follows: conceptualization, S.L. and X.H.; methodology, A.R.S.K., Y.J., and X.H.; software, S.L; writing (original draft preparation), A.R.S.K. and X.H; writing (review and editing), Y.J., S.L., and X.H.; project administration, X.H.; funding acquisition, X.H.

We declare no conflicts of interest.

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
