## [Reviewer comments · Microbiology Spectrum]

Microbiology Spectrum

***Aspergillus nidulans* AmyG functions as an intracellular α -amylase to promote alpha-glucan synthesis**

Xiaoxiao He, Alia Kazim, Shengnan Li, and Yuting Jiang

Corresponding Author(s): Xiaoxiao He, Northeast Normal University

Review Timeline:

Submission Date:	July 13, 2021
Editorial Decision:	August 26, 2021
Revision Received:	October 15, 2021
Accepted:	October 19, 2021

Editor: Gustavo Goldman

Reviewer(s): The reviewers have opted to remain anonymous.

Transaction Report:

DOI: <https://doi.org/10.1128/Spectrum.00644-21>

August 26, 2021

Dr. Xiaoxiao He
Northeast Normal University
5268 Renmin Avenue
Changchun
China

Re: Spectrum00644-21 (*Aspergillus nidulans* AmyG functions as an intracellular α -amylase to promote alpha-glucan synthesis)

Dear Dr. Xiaoxiao He:

Thank you for submitting your manuscript to Microbiology Spectrum. When submitting the revised version of your paper, please provide (1) point-by-point responses to the issues raised by the reviewers as file type "Response to Reviewers," not in your cover letter, and (2) a PDF file that indicates the changes from the original submission (by highlighting or underlining the changes) as file type "Marked Up Manuscript - For Review Only". Please use this link to submit your revised manuscript - we strongly recommend that you submit your paper within the next 60 days or reach out to me. Detailed information on submitting your revised paper are below.

Link Not Available

Sincerely,

Gustavo Goldman

Journals Department
Reviewer comments:

Reviewer #1 (Comments for the Author):

In this manuscript, the authors have studied AmyG function to be an intracellular alpha-amylase promoting alpha-glucan synthesis in *Aspergillus nidulans*. There are several points that are not clear in the manuscript:

1. I have a major issue with alpha-glucan isolation and quantification: (i) could the authors provide any reference protocol for alpha-glucan separation? (ii) in the protocol, they disrupt the cells to collect cell wall, which is then washed and freeze-dried; fungal cell wall is known to harbor proteins, and most of them are cell-wall modifying enzymes, but there was no step included to remove these cell wall bound proteins prior to NaOH treatment, (iii) how the authors are sure that the transparent pellet contains alpha-glucan? (iv) anthrone assay is not specific for alpha-glucan, but generally used to test carbohydrates, (v) can the authors provide evidence for the extracted material is only alpha-glucan?

2. Ideal IPTG concentration for protein expression has already been tested and found to be around 1 mM; did the authors test other growth temperatures for AmyG expression than 37 and 16°C?

3. Why specifically maltose or maltotriose was used? Did the authors try other oligosaccharides, specifically alpha-1,3-linked oligosaccharides? Moreover, the assay duration is 3 h; did they try prolonging the assay duration? Because glucanotransferase activity is known to start with hydrolytic process, turning into transferase activity in the due course.

4. In amyG deletion mutant, addition of maltose/maltotriose could be restoring osmotic balance to regain normal growth; authors need to check it, using alternative components (e.g., sorbitol) in the culture medium.

Minor points:

5. Introduction, the authors need to be careful with the sentences, for e.g., alpha-glucan masks beta-glucan from host immune system and alpha-glucan synthase mutant becomes avirulent

6. Line 54, correct as 'cannot be produced by'

7. Figure numbers are not associated with the figures presented

Reviewer #2 (Comments for the Author):

The authors claimed that AmyG (an intracellular amylase) is the source of maltose/maltotriose for cell wall alpha-glucan synthesis. Which would be the substrate in vivo? Glycogen? Is the AmyG active on glycogen? Despite it is a piece of evidence, it is still an incomplete piece of evidence.

Section - Enzymatic characterization of AmyG

Overall the enzymatic characterization presented in the manuscript is preliminary. Various external factors can influence enzymatic kinetics such as pH, temperature, enzyme concentration, time of reaction, substrate concentration, etc. However, the AmyG enzymatic reactions were carried out exclusively for 3h, pH 6.5 at 37{degree sign}C. Indeed, the paradox hydrolysis x synthesis can even be dependent on the previously mentioned external factors. Moreover, how is the enzyme concentration (mM) in 10 μ L of AmyG (in 25% glycerol-PBS)??

Both Figure 3B and Figure 4 show starch hydrolysis using a commercial α -amylase and AmyG. However, the G4 and the smear probably representing G5/G6/G7 (Figure 3B, sample 2) were not detected by HPLC (Figure 4, sample 2)? Moreover, G2 and G3 were the major products from α -amylase reactions (Figure 4, sample 1), instead of G1 and G2 as mentioned. Finally, the authors claimed that G2 and G3 are the major products from starch hydrolysis by AmyG, however assuming Figure 4B, I would say G1 and G3 were produced at similar levels.

I strongly suggest the authors use the powerful HPLC tool to quantify each product released by both enzymes. In addition, the authors would add data on AmyG kinetics parameters such as Kcat/Km to demonstrate the catalytic efficiency of the enzyme on different substrates, since the major claim in the Introduction section (li.60-61) is "Previous studies have shown the effect of AmyG deletion on the cell wall of *A. nidulans* but no in vitro characterization and definite function has been reported."

Additional comments

li.54. by produced??

li.192. maltotriosen

li.201-202: These data suggested that AmyG hydrolyzes starch producing maltose and maltotriose as major products.

li.206-207: typo and punctuation issue.

li.207: disaccharide/oligosaccharide = saccharides

li.225-226: "Moreover, extra maltotriose in medium also showed an inhibitory effect on α -glucan synthesis (Fig. 5B)." I've expected results with different G3 concentrations.

Figure 2. concatenations.

Figure S2. The cell lysate from non-amyG express *E. coli* cells

Figure 3. Missing control. A control reaction using 50 min boiled enzyme should be added to the experiment.

Figure 5. Incomplete experiment. This experiment clearly showing that G2/G3 supplementation rescue the amyG phenotype, must be accompanied by the quantification of α -1,3-glucan concentration of agsB Δ strain. I also wondered how about the phenotype of OE_amyG/agsB Δ and OE_agsB/amyG Δ strains supplemented with G2 and G3.

Staff Comments:

Preparing Revision Guidelines

Please return the manuscript within 60 days; if you cannot complete the modification within this time period, please contact me. If you do not wish to modify the manuscript and prefer to submit it to another journal, please notify me of your decision immediately so that the manuscript may be formally withdrawn from consideration by Microbiology Spectrum.

If you would like to submit an image for consideration as the Featured Image for an issue, please contact Spectrum staff.

Reviewer comments:

Reviewer #1 (Comments for the Author):

In this manuscript, the authors have studied AmyG function to be an intracellular alpha-amylase promoting alpha-glucan synthesis in *Aspergillus nidulans*. There are several points that are not clear in the manuscript:

1. I have a major issue with alpha-glucan isolation and quantification: (i) could the authors provide any reference protocol for alpha-glucan separation?(ii) in the protocol, they disrupt the cells to collect cell wall, which is then washed and freeze-dried; fungal cell wall is known to harbor proteins, and most of them are cell-wall modifying enzymes, but there was no step included to remove these cell wall bound proteins prior to NaOH treatment, (iii) how the authors are sure that the transparent pellet contains alpha-glucan? (iv) anthrone assay is not specific for alpha-glucan, but generally used to test carbohydrates, (v) can the authors provide evidence for the extracted material is only alpha-glucan?

Response: We thank the reviewer to point out these issues. The strategy for cell wall preparation was adopted from Momany et al.(2004) and the alpha-glucan quantification was adopted from Marion et al. (2006), which used anthrone assay to quantify the glucose content in the alkaline soluble fraction. We have previously verified this strategy in *A.nidulans* using an *agsB* deletion strain (He et al., 2014), which had almost no glucose left in the alkaline soluble fraction. Moreover, the *agsB* overexpression strain (He et al., 2014) had significantly more alkaline soluble fraction (the transparent pellet) and correspondingly more glucose. We also used immuno-staining method to examine the alpha-glucan content in these two strains (He et al., 2014). Results from both methods perfectly match each other. These data suggested that the glucose content in the alkaline soluble fraction was mostly from alpha-glucan. In addition, result from other publications also supported this conclusion. Yoshimi et al. (2013) used similar strategy to isolate alpha-glucan from *A. nidulans* cell wall by alkaline, but quantify this fraction using high-performance anion-exchange chromatography (HPAC). Their results also proved that the glucose content in the alkaline soluble fraction was mostly from alpha-glucan. Thus, all strategies for alpha-glucan quantification were based on isolation of alpha-glucan using alkaline solution and further quantify the glucose content. So far there is no data implying that any cell wall harboring protein may interfere the cell wall isolation and quantification process.

References:

Marion,C.L.,Rappleye,C.A.,Engle,J.T.,andGoldman,W.E.2006. An alpha-(1,4)-amylase is essential for alpha-(1,3)-glucan production and virulence in *Histoplasma capsulatum*. *Mol Microbiol* 62:970–983.

Momany, M., Lindsey, R., Hill, T.W., Richardson, E.A., Momany, C., Pedreira, M., et al.

2004. The *Aspergillus fumigatus* cell wall is organized in domains that are remodeled during polarity establishment. *Microbiology* 150:3261–3268

Yoshimi A, Sano M, Inaba A, Kokubun Y, Fujioka T, Mizutani O, Hagiwara D, Fujikawa T, Nishimura M, Yano S, Kasahara S, Shimizu K, Yamaguchi M, Kawakami K, Abe K. 2013. Functional analysis of the α -1,3-glucan synthase genes *agsA* and *agsB* in *Aspergillus nidulans*: AgsB is the major α -1,3-glucan synthase in this fungus. *PLoS One* 8:e54893.

He XX, Li SN, Kaminskyj SGW. 2014. Characterization of *Aspergillus nidulans* α -glucan synthesis: roles for two synthases and two amylases. *Mol Microbiol* 91:579-95.

2. Ideal IPTG concentration for protein expression has already been test and found to be around 1 mM; did the authors test other growth temperatures for AmyG expression than 37 and 16oC?

Response: We initially tried to induced AmyG expression at 37 °C. However, we did not find significant amount of AmyG in the soluble fraction of cell lysate. These results could be found in the new figure S1A.

3. Why specifically maltose or maltotriose was used? Did the authors try other oligosaccharides, specifically alpha-1,3-linked oligosaccharides? Moreover, the assay duration is 3 h; did they try prolonging the assay duration? Because glucanotransferase activity is known to start with hydrolytic process, turning into transferase activity in the due course.

Response: The starch that used in this study is linear 1,4-linked glucose. The expected outcomes from degradation should be alpha-1,4-linked oligosaccharides. In addition, the sequence analysis showed AmyG belongs to GH13_5, which should have the ability to degrade 1,4-linked glucose but not 1,3-linked glucose. We previously characterized 1,3-glucanase in *A. nidulans*, which showed strong ability to degrade alpha-glucan *in vivo* (He et al., 2017). The sequences of glucanase (AgnB and MutA, which belong to GH71 family) are quite different from AmyG. So we did not test if AmyG has the ability to degrade alpha-1,3-linked oligosaccharides. In addition, we tried to hydrolyze glycogen using AmyG during the revision. However, results showed AmyG could not hydrolyze glycogen at all (see revised Fig. 3C).

As the reviewer suggested, we tried to prolong the duration of the glucanotransferase assay to 12 h. However, the same results were found for these two assays with different time duration. Please find the new result in Fig. 3B.

Reference:

He XX, Li SN, Kaminskyj SGW. 2017. An Amylase-Like Protein, AmyD, Is the Major Negative Regulator for α -Glucan Synthesis in *Aspergillus nidulans* during the Asexual Life Cycle. *International Journal of Molecular Sciences*, 27 Mar 2017, 18(4)

4. In amyG deletion mutant, addition of maltose/maltotriose could be restoring osmotic balance to regain normal growth; authors need to check it, using alternative components (e.g., sorbitol) in the culture medium.

Response: The osmotic balance indeed is an important factor that regulates cell growth. As the reviewer suggested, we tried to compensate the phenotypic defect of amyG KO strain using extra glucose, sucrose and sorbitol in the medium. Results showed none of them improved the tiny colony formation in the shaken liquid medium. Thus, the restoring of amyG KO strain should be not due to change of osmotic balance. These results was added in the revised manuscript. Please see Fig. S5.

Minor points:

5. Introduction, the authors need to be careful with the sentences, for e.g., alpha-glucan masks beta-glucan from host immune system and alpha-glucan synthase mutant becomes avirulent

Response: As the reviewer suggested, we tuned down the expression of these sentences. We think we faithfully followed the idea of original publications. Please see lines 43-47 for detail.

Line 54, correct as 'cannot be produced by'

Response: We thank the reviewer to point out this mistake. This was corrected as suggested. Please see line 55 for detail.

7. Figure numbers are not associated with the figures presented

Response: We have re-organized some figures and checked this problem throughout the manuscript.

Reviewer #2 (Comments for the Author):

The authors claimed that AmyG (an intracellular amylase) is the source of maltose/maltotriose for cell wall alpha-glucan synthesis. Which would be the substrate *in vivo*? Glycogen? Is the AmyG active on glycogen? Despite it is a piece of evidence, it is still an incomplete piece of evidence.

Response: During the revision of this manuscript. We found AmyG could not hydrolyze glycogen (see revised Fig. 3C), which is similar as a previous report (Camacho et al., 2012). Thus, the *in vivo* substrate is still unknown.

Reference:

Camacho, E.; Sepulveda, V. E.; Goldman, W. E.; San-Blas, G.; Nino-Vega, G. A., Expression of *Paracoccidioides brasiliensis* AMY1 in a *Histoplasma capsulatum* amy1 Mutant, Relates an alpha-(1,4)-Amylase to Cell Wall alpha-(1,3)-Glucan Synthesis. *Plos One* 2012, 7 (11). DOI: 10.1371/journal.pone.0050201.

Section - Enzymatic characterization of AmyG

Overall the enzymatic characterization presented in the manuscript is preliminary. Various external factors can influence enzymatic kinetics such as pH, temperature, enzyme concentration, time of reaction, substrate concentration, etc. However, the AmyG enzymatic reactions were carried out exclusively for 3h, pH 6.5 at 37°C. Indeed, the starch hydrolysis x synthesis can even be dependent on the previously mentioned external factors. Moreover, how is the enzyme concentration (mM) in 10 µL of AmyG (in 25% glycerol-PBS)??

Response: As the reviewer suggested, we tested the hydrolytic activity of AmyG at different PH (4.5-8.5) and temperature (20-37°C). During the revision of this manuscript, we further modify the starch hydrolysis reaction. Variations of starch concentration, time of reaction and enzyme concentration were all considered and described in the Method section. The new results were included in the revised manuscript. Please see Fig. 3, S2 and S3 for detail.

In our modified reaction, we used 5 µL of freshly purified AmyG, which normally contains 3 µg of enzyme. Such information was added in the revised manuscript.

Both Figure 3B and Figure 4 show starch hydrolysis using a commercial α -amylase and AmyG. However, the G4 and the smear probably representing G5/G6/G7 (Figure 3B, sample 2) were not detected by HPLC (Figure 4, sample 2)? Moreover, G2 and G3 were the major products from α -amylase reactions (Figure 4, sample 1), instead of G1 and G2 as mentioned. Finally, the authors claimed that G2 and G3 are the major products from starch hydrolysis by AmyG, however assuming Figure 4B, I would say G1 and G3 were produced at similar levels.

I strongly suggest the authors use the powerful HPLC tool to quantify each product released by both enzymes. In addition, the authors would add data on AmyG kinetics parameters such as K_{cat}/K_m to demonstrate the catalytic efficiency of the enzyme on different substrates, since the major claim in the Introduction section (li.60-61) is "Previous studies have shown the effect of AmyG deletion on the cell wall of *A. nidulans* but no in vitro characterization and definite function has been reported."

Response: During the revision of this manuscript, we re-performed the most of the TLC analysis. As the reviewer suggested, there were small amount of G4-G6 oligosaccharides presented in the starch hydrolysis reaction, which is different from the final products of *A. oryzae* amylase. We used to HPAEC-PAD to identify such products and confirmed the presence of G4-G6 oligosaccharides. In addition, we also found AmyG could have a weak glucanotransferase activity, and this activity is quite specific to use maltotriose as the substrate. When maltotriose was incubated with AmyG, it will formed such G4-G6 oligosaccharides, but the formation of such compounds maintained at very low level even for a prolonged reaction period (12 h). Thus, we have changed our conclusion as "AmyG has strong hydrolysis but weak glucanotransferase activity, which produced maltose as the major prouduct, which G3-G6 oligosaccharides were also presented". Please see lines 210-246 for detail.

As the reviewer suggested, we further quantified the enzyme kinetics parameters. Since the HPLC machine was not easily available at my current institute, we used the

bicinchoninic acid method to quantify the reducing sugar ends. We focused on starch hydrolysis ability. Our results showed the AmyG activity on starch was $67.4 \pm 7.3 \mu\text{mol}^{-1}$ reducing ends $\text{mg}^{-1} \text{min}^{-1}$. The K_m for starch was around 0.12%~0.16% w/v.

Additional comments

li.54. by produced??

Response: Corrected as “be produced”. Please see lines 55 for detail.

li.192. maltotriosen

Response: Corrected as “maltotriose”.

li.201-202: These data suggested that AmyG hydrolyzes starch producing maltose and maltotriose as major products.

Response: Corrected as suggested. Please see lines 245-246 for detail.

li.206-207: typo and punctuation issue.

Response: Corrected. Please see lines 249-252 for detail.

li.207: disaccharide/oligosaccharide = saccharides

Response: Corrected as suggested.

li.225-226: "Moreover, extra maltotriose in medium also showed an inhibitory effect on α -glucan synthesis (Fig. 5B)." I've expected results with different G3 concentrations.

Response: During the revision, we did try to grow A1149 in different maltotriose containing medium (ranging from 0.25% to 2%). However, in 0.5% and above concentrations the α -glucan content did not further decreased. Therefore, this inhibitory effect was not quite concentration dependent. To not confound our current conclusion, we did not added such data in the revised manuscript.

Figure 2. concatenations.

Response: Corrected as “concentrations”.

Figure S2. The cell lysate from non-amyG express E. coli cells

Response: Corrected as suggested.

Figure 3. Missing control. A control reaction using 50 min boiled enzyme should be added to the experiment.

Response: As the reviewer suggested, we added the boiled inactive enzyme control. As expected, the boiled enzyme cannot hydrolyze starch at all. Please see the revised manuscript Fig. 3A for detail.

Figure 5. Incomplete experiment. This experiment clearly showing that G2/G3 supplementation rescue the amyG phenotype, must be accompanied by the quantification of α -1,3-glucan concentration of agsB Δ strain. I also wondered how about the phenotype of OE_amyG/agsB Δ and OE_agsB/amyG Δ strains supplemented

with G2 and G3.

Response: We further performed alpha-glucan quantification experiment in *agsB* Δ strain. Result showed the addition of maltose or maltotriose did not alter the alpha-glucan content in *agsB* Δ strain. The new result was added in the revised Fig. 5B. For testing of OE_*amyG*/*agsB* Δ and OE_*agsB*/*amyG* Δ strains, we did not have these strains in stock at the current institute. Our previous results showed that AgsB is the rate-limiting enzyme in alpha-glucan synthesis, and its function was dependent on AmyG. The phenotype of OE_*amyG*/*agsB* Δ would be the same as *agsB* Δ strain, and the maltose should be able to compensate the phenotypic defect in OE_*agsB*/*amyG* Δ . We would like to test these hypothesis in the future.

October 19, 2021

Dr. Xiaoxiao He
Northeast Normal University
5268 Renmin Avenue
Changchun
China

Re: Spectrum00644-21R1 (*Aspergillus nidulans* AmyG functions as an intracellular α -amylase to promote alpha-glucan synthesis)

Dear Dr. Xiaoxiao He:

The authors have addressed all the comments and suggestions of the reviewers and I believe the manuscript is ready to be accepted.

Your manuscript has been accepted, and I am forwarding it to the ASM Journals Department for publication. You will be notified when your proofs are ready to be viewed.

Sincerely,

Gustavo Goldman
Editor, Microbiology Spectrum
